# Genome-wide meta-analysis of phytosterols reveals five novel loci and a detrimental effect on coronary atherosclerosis

Markus Scholz [1,2✉], Katrin Horn [1,2], Janne Pott [1,2], Arnd Gross [1], Marcus E. Kleber [3], Graciela E. Delgado[3], Pashupati Prasad Mishra [4], Holger Kirsten [1,2], Christian Gieger [5,6,7], Martina Müller-Nurasyid[8,9,10,11], Anke Tönjes[12], Peter Kovacs [7,12], Terho Lehtimäki [4], Olli Raitakari[13], Mika Kähönen[13], Helena Gylling[14], Ronny Baber[2,15], Berend Isermann[15], Michael Stumvoll [12], Markus Loeffler[1,2], Winfried März[3,16,17], Thomas Meitinger[18], Annette Peters [19], Joachim Thiery[2,15], Daniel Teupser[20] & Uta Ceglarek[2,15]

Phytosterol serum concentrations are under tight genetic control. The relationship between phytosterols and coronary artery disease (CAD) is controversially discussed. We perform a genome-wide meta-analysis of 32 phytosterol traits reflecting resorption, cholesterol synthesis and esterification in six studies with up to 9758 subjects and detect ten independent genome-wide significant SNPs at seven genomic loci. We confirm previously established associations at *ABCG5/8* and *ABO* and demonstrate an extended locus heterogeneity at *ABCG5/8* with different functional mechanisms. New loci comprise *HMGCR*, *NPC1L1*, *PNLIPRP2*, *SCARB1* and *APOE*. Based on these results, we perform Mendelian Randomization analyses (MR) revealing a risk-increasing causal relationship of sitosterol serum concentrations and CAD, which is partly mediated by cholesterol. Here we report that phytosterols are polygenic traits. MR add evidence of both, direct and indirect causal effects of sitosterol on CAD.

[1] Institute for Medical Informatics, Statistics and Epidemiology, Medical Faculty, University of Leipzig, Leipzig, Germany. [2] LIFE Research Center for Civilization Diseases, Medical Faculty, University of Leipzig, Leipzig, Germany. [3] Medical Faculty Mannheim, Vth Department of Medicine, Heidelberg University, Heidelberg, Germany. [4] Department of Clinical Chemistry, Fimlab Laboratories, and Finnish Cardiovascular Research Center - Tampere, Faculty of Medicine and Health Technology, Tampere University, Tampere 33520, Finland. [5] Research Unit of Molecular Epidemiology, Helmholtz Zentrum München, German Research Center for Environmental Health, Munich, Germany. [6] Institute of Epidemiology, Helmholtz Zentrum München, German Research Center for Environmental Health, Munich, Germany. [7] German Center for Diabetes Research (DZD), Munich, Germany. [8] Institute of Genetic Epidemiology, Helmholtz Zentrum München - German Research Center for Environmental Health, 85764 Neuherberg, Germany. [9] Institute for Medical Information Processing, Biometry and Epidemiology, Medical Faculty, Ludwig-Maximilians University of Munich, Munich, Germany. [10] Institute of Medical Biostatistics, Epidemiology and Informatics (IMBEI), University Medical Center, Johannes Gutenberg University, 55101 Mainz, Germany. [11] Department of Internal Medicine I (Cardiology), Hospital of the Ludwig-Maximilians-University Munich, 81377 Munich, Germany. [12] Medical Department III – Endocrinology, Nephrology, Rheumatology, University Hospital Leipzig, Leipzig, Germany. [13] Department of Clinical Physiology, Tampere University Hospital, and Finnish Cardiovascular Research Center - Tampere, Faculty of Medicine and Health Technology, Tampere University, Tampere 33521, Finland. [14] Heart and Lung Center, Cardiology, University of Helsinki and Helsinki University Hospital, Helsinki, Finland. [15] Institute for Laboratory Medicine Clinical Chemistry and Molecular Diagnostics, University Hospital Leipzig, Leipzig, Germany. [16] Synlab Academy, Synlab Holding Deutschland GmbH, Augsburg, Germany. [17] Clinical Institute of Medical and Chemical Laboratory Diagnostics, Medical University Graz, Graz, Austria. [18] Institue for Human Genetics, Technical University of Munich, Munich, Germany. [19] Institue of Epidemiology, Helmholtz Zentrum München, German Research Center for Environmental Health, Munich, Germany. [20] Insitute for Laboratory Medicine, Ludwig-Maximilians University of Munich, Munich, Germany. ✉email: markus.scholz@imise.uni-leipzig.de

Phytosterols are cholesterol homologues synthesized by plants only. Therefore, phytosterol concentrations in mammals can be completely attributed to nutrition, where it can mainly be found in plant oils, nuts and seeds[1–3]. Normal diet contains approximately equal molar amounts of cholesterol and phytosterols, but, serum levels of phytosterols are kept ~200-fold lower compared to cholesterol[4,5]. A heterodimeric ATP-dependent transmembrane complex consisting of ABCG5 and ABCG8 hemi-transporters, expressed in intestine and liver, plays a key role in the excretion of sterols, keeping serum phytosterol concentrations low[6].

Phytosterol concentrations are known to be under tight genetic control[7]. Pathologically increased serum concentrations of phytosterols resulting from loss of function mutations of *ABCG5/G8* are described resulting in the severe condition of phytosterolemia, most prominently, sitosterolemia[8]. In a previous work, we identified genetic factors responsible for regulating serum phytosterol levels at physiological levels in the general population[9]. In this genome-wide association study (GWAS), we identified three independent common SNPs associated with phytosterols, two in *ACBG8* and one in *ABO*. This study was performed with a limited sample size of 1495 subjects and replication in 2917 subjects. Still, until now, this has been the only GWAS of this phenotype.

Importantly, a certain level of phytosterols is discussed to be beneficial, as the cholesterol-lowering effect of phytosterol supplementation is well established (e.g.[10,11], see ref. [3] for a recent summary). A consumption of 1–2 g/day was shown to lower low-dense lipoprotein-cholesterol (LDL-C) plasma concentration by about 5–16%[12]. Several physiological explanations of this phenomenon were proposed, comprising competitive incorporation of cholesterol and phytosterols into micelles[13] and a multitude of molecular regulatory processes of phytosterols on genes involved in cholesterol homeostasis[14].

Despite its cholesterol-lowering effect, the relationship between serum phytosterol concentrations and coronary artery disease (CAD) risk is conversely debated. Experimental studies of phytosterol-enriched diet in mice showed that phytosterols are atherogenic[15,16]. But other animal studies could not find such effects or even the opposite[10,17]. In humans, the Mendelian disorder sitosterolemia is associated with an increased risk for atherosclerosis[18]. Phytosterols were found to be accumulated in carotid plaques[19,20] and longitudinal epidemiologic studies identified serum phytosterols as risk factors of subsequent cardiovascular events (e.g. ref. [21] others summarized in ref. [22]). In line with these observations, in our former study, we found that all three variants associated with higher phytosterol levels were also associated with increased CAD risk[9]. On the other hand, a meta-analysis could not find any direct associations of phytosterols with CAD risk[23] and the authors attributed cholestanol-to-cholesterol ratio as the causal factor driving the association seen in our former study[24]. A recent review[25] disputes this interpretation by pointing out that the genetic effect size of the ABCG5/8 locus on cholesterol traits is much smaller than on phytosterols and that there is no evidence of a beneficial effect of phytosterol supplementation regarding CAD risk despite of a clear improvement of cholesterol parameters. A stringent Mendelian randomization analysis of that issue was not performed so far.

These contradicting findings could be attributed to the small sample size of human studies[2], tissue and phytosterol species-specific effects[26] and the close and complex interactions of phytosterols and cholesterol on several molecular levels[14] making it difficult to separate the effects of phytosterols and cholesterol. Moreover, studies considering phytosterols as serum biomarkers often found a positive correlation with CAD endpoints[22] while phytosterol supplementation as nutrition intervention more often find negative correlations[27,28].

We here presented the results of a meta-GWAS in a significantly larger sample of up to 9758 individuals from six studies to gain a deeper insight into the genetics of phytosterol metabolism by identifying additional genetic factors responsible for regulating serum phytosterol concentrations. We also analysed genetics of free and esterified phytosterol species as well as ratios of free to esterified phytosterols and of phytosterols to cholesterol or lanosterol. Based on our findings, we aimed at unravelling the causal relationships of phytosterols, cholesterol and CAD by performing a stringent multi-instrument Mendelian randomization analysis.

## Results

**Sterol clustering and correlation**. Applying hierarchical clustering of the phytosterol traits revealed that absolute phytosterol serum concentrations are closely correlated. Ratios to lanosterol and free to esterified ratios are clearly separated. Clustering and correlation heatmap are presented in Supplementary Fig. S1.

**Results of meta-GWAS**. Meta-GWAS results of 32 phytosterol traits and ratios showed no signs of genomic inflation in fixed effect modelling (maximum Lambda of 1.019 for the ratio of free brassicasterol to lanosterol, see Supplementary Data S4). Results of random effects modelling are clearly deflated as expected and is provided as secondary statistics.

A total of 584 SNPs distributed over seven different genomic loci showed genome-wide significance with at least one of the phytosterol traits. A circular plot comparing single trait vs. ratio-based associations is shown in Fig. 1. Regional association plots of the seven loci with genome-wide significant hits are shown as Supplementary Fig. S2 (2p21 locus, showing conditional results) and S3 (other loci). Results of the fine-mapping of the 2p21 locus are depicted in Fig. 2.

Five of the genome-wide significant hits are observed for both, absolute phytosterols and phytosterols to cholesterol or lanosterol ratios. One hit is only observed for absolute phytosterol levels while another one was only observed for phytosterol to lanosterol ratios. Best associated traits do not comprise free to esterified phytosterols, i.e. these traits did not contribute to hit discovery. Statistics of all genome-wide significant SNPs and their annotations are provided in Supplementary Data S5.

Cojo-Select analysis revealed four independent signals at the 2p21 locus, while for the other loci, no other independent signals were found. Basic characteristics of the resulting ten independent SNPs are shown in Table 1.

Forest plots of the ten SNPs are provided in Supplementary Fig. S4 and show direction consistency of the studies in all but one situation (concerning YFS with the smallest sample size). Most of the SNPs are associated with more than one phytosterol trait, consistent with the observed correlation structure between traits. Co-associations in relation to the hierarchical clustering of the traits are depicted in Fig. 3, corresponding numerical values can be found in Supplementary Data S6. Comparison of beta estimates for the different absolute phytosterol traits (free, esterified, total) revealed strong similarity (see Supplementary Fig. S5A). There are also such similarities between total phytosterols and their ratios with total cholesterol or free lanosterol with two exceptions, namely 2p21 and 5q13.3 as discussed below (see Supplementary Fig. S5B).

Since the ten independent SNPs do not always show the strongest associations for each trait per locus, we also present locus-wide top-associations in Supplementary Data S7.

Credible sets per independent variant and corresponding annotations are provided in Supplementary Data S8. Colocalization analyses were performed regarding LDL-C, total cholesterol,

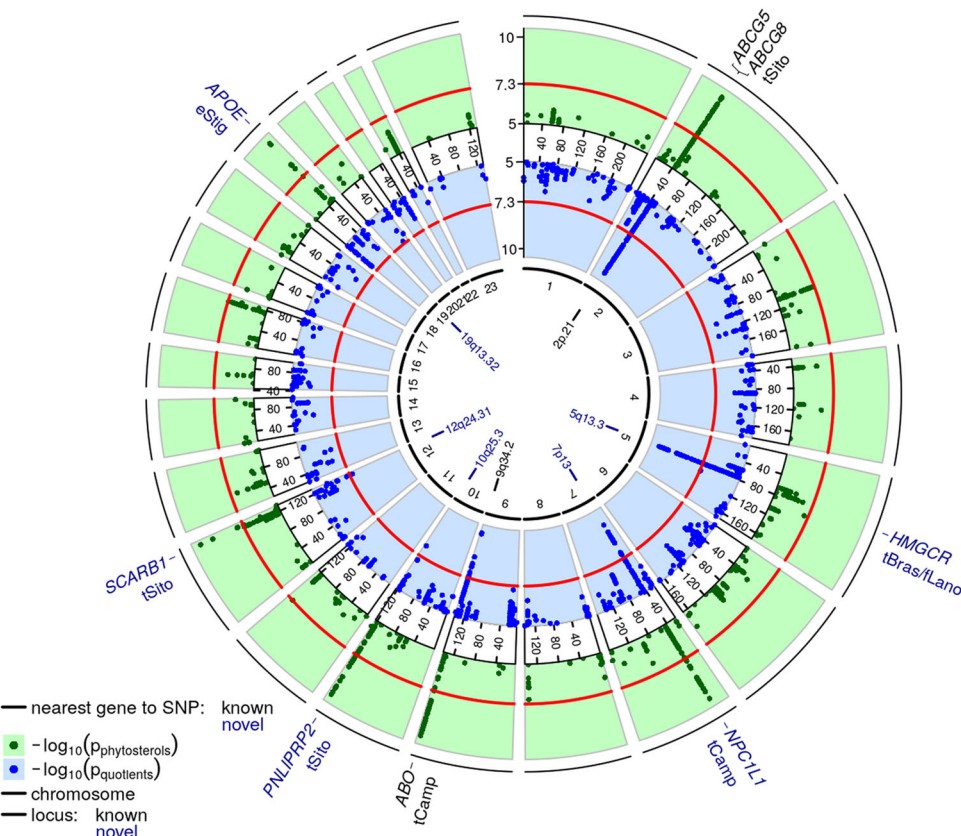

**Fig. 1 Circus plot of genetic associations of phytosterol traits.** We present results of our genome-wide association analyses as circus plot. Dots in green and blue rings correspond to association statistics ($-\log_{10}$(p-values) of fixed effect meta-analysis of gene-dose effects) of raw phytosterols, respectively normalized phytosterols and quotients. Only values larger than five are displayed. Values larger than ten are set to ten. Slices correspond to chromosomes. Physical positions are shown in Mb. The red circles mark the level of genome-wide significance ($-\log_{10}(5 \times 10^{-8})$). In the centre of the plot, new loci are shown in blue colour, while known loci are depicted in black. Most plausible candidate genes per locus and respective best associated traits are provided at the outer ring. Abbreviations of traits are given in Supplementary Data S18.

CAD and eQTLs of derived candidate genes in eight tissues. Major results are summarized in Fig. 4. Colocalization with other candidate eQTLs is presented in Supplementary Fig. S6. Numerical results are presented in Supplementary Data S9.

The seven genome-wide significant loci comprise the two associations already discovered by our previous single study GWAS[9] but with other top-hits due to our denser marker map. Thus, five loci (5q13.3, 7p13, 10q25.3, 12q24.31, 19p13.3) are considered novel.

We first characterize additional results of the known loci.

**Known loci**

*2p21.* At 2p21, we confirm the observed strong association with all phytosterols. The formerly found rs4245791 was tagged by the new top-hit rs4299376 ($r^2 = 0.97$, $p = 1.5 \times 10^{-151}$ in unconditional analysis of the top-associated trait total sitosterol, $p = 9.5 \times 10^{-74}$ in conditional analysis). The conditional 99% credible set contains three SNPs in high LD with the lead variant ($r^2 > 0.96$) including rs4245791. The conditional statistics of the new lead SNP rs4299376 colocalizes with an eQTL of *ABCG8* in colon tissue[29] (PP4 = 99.7%) and also with CAD (PP4 = 98.8%) but interestingly, not with cholesterol (PP3 = 99.7%, see Fig. 4).

Conditional analysis revealed four independent associations for that locus (see Fig. 2). The second strongest independent association was observed for rs11887534, which is in LD with our previously reported variant rs41360247 ($r^2 = 0.93$, $p = 8.3 \times 10^{-39}$ in conditional analysis). The 99% credible set contains seven variants in high LD with the lead variant

($r^2 > 0.93$) including rs41360247. Rs11887534 displays a strong deleteriousness score (CADD = 22.7). The minor allele represents a well-known non-synonymous coding mutation of ABCG8 (D19H), which results in lower phytosterol levels due to a gain of function mutation[30]. Colocalizations of this locus were observed with cholesterol (PP4 = 94.7%) and CAD (PP4 = 97.2%) but no eQTLs. Though, the signal for cholesterol is clearly weaker than for total sitosterol in terms of explained variance (0.2% for total cholesterol, 1.7% for total sitosterol).

A third independent association was observed for rs7598542 ($5.1 \times 10^{-10}$ in conditional analysis). This variant lies in a common haploblock with the two strongest associations (see Fig. 2). Colocalization analysis revealed co-associations of this locus with CAD (PP4 = 99.6%), and weakly, with an *ABCG8* eQTL in colon tissue (PP4 = 56%). The 99% credible set contains 16 variants. Among those, rs4148217 showed the highest CADD score of 14.8, since this variant is again a non-synonymous coding mutation of *ABCG8* (T400K), which however, is considered benign. Thus, it is not clear whether this association is driven by gene regulation or protein function or both.

A fourth independent association was found for rs78451356 outside of the haploblock of the three variants above and in close proximity to *ABCG5* ($1.1 \times 10^{-14}$). The 99% credible set contains 12 variants, all in high LD ($r^2 > 0.89$) with the lead variant. The strongest CADD score of 13.0 corresponds to rs8302 which is an intron variant of *ABCG5* and in the 3'UTR of *DYNC2LI1*. But, no colocalizations of our phytosterol associations were observed with respective eQTLs.

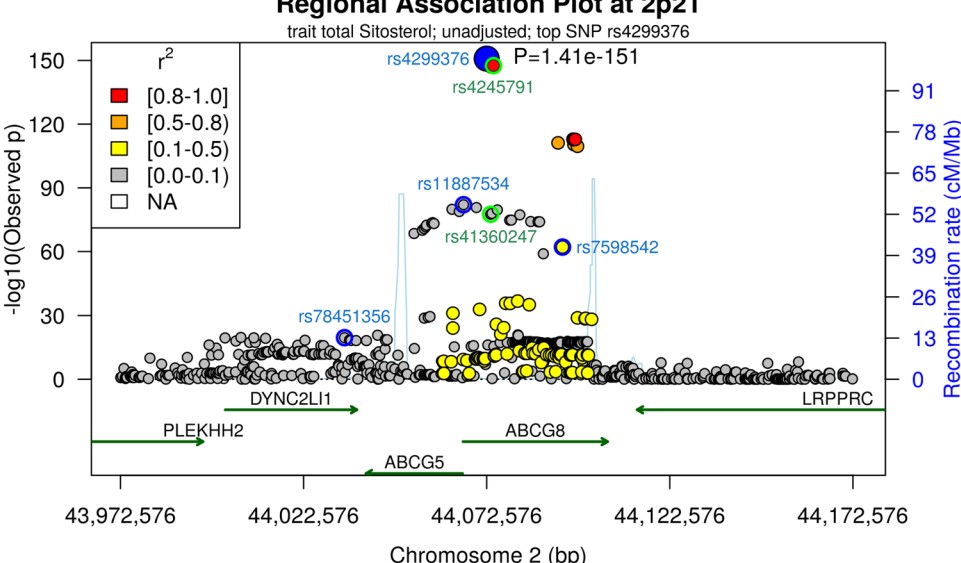

**Fig. 2 Fine-mapping of 2p21 locus.** We present a regional association plot of the 2p21 region. A window of 100 kb around the top-SNP is presented. Dots correspond to SNP position and $-\log_{10}$(p-values) of fixed effect meta-analysis of gene-dose effects for total sitosterol. The large blue dot depicts the top-SNP of that locus. Colours of small dots indicate LD ($r^2$) with the top-SNP. We also provide annotated genes within the locus and recombination rates to mark haplo-blocks. We present unconditioned association results. According to Cojo-select analysis, four independent variants are detected (blue circles plus the main hit). Of note, three of the variants are confined within one haplo-block and could be attributed to *ABCG8* while the fourth variant (rs78451356) lies in a neighbouring haplo-block corresponding to *ABCG5/DYNC2LI1*. Green circles mark variants reported in our former GWAS. Regional association plots of the respective conditional association results are provided as Supplementary Fig. S2.

**Table 1 Description of the ten independent genome-wide significant variants identified in our GWAS).**

| Cytoband | Best trait | Independent SNP rsID | #SNPs in Credible set 99% (95%) | Candidate gene (kb) | Effect allele/ other allele | Weighted effect allele freq | $I^2$ (%) | (Cond.) Beta | (Cond.) p-val | Exp. Var (%) |
|---|---|---|---|---|---|---|---|---|---|---|
| 2p21 | tSito | rs4299376 | 3 (3) | *ABCG8* (0) | T/G | 0.68 | 0 | −0.133 | $9.5 \times 10^{-74}$ | 3.27 |
| 2p21 | tSito | rs11887534 | 7 (5) | *ABCG8* (0) | C/G | 0.065 | 0 | −0.178 | $8.3 \times 10^{-39}$ | 1.71 |
| 2p21 | tSito | rs7598542 | 16 (7) | *ABCG8* (0) | C/G | 0.21 | 52 | −0.052 | $5.1 \times 10^{-10}$ | 0.39 |
| 2p21 | tSito | rs78451356 | 12 (11) | *ABCG5* (5.9) | G/T | 0.17 | 9.4 | 0.069 | $1.1 \times 10^{-14}$ | 0.61 |
| 5q13.3 | tBras/fLano | rs12916 | 37 (21) | *HMGCR* (0) | C/T | 0.42 | 0 | −0.059 | $2.3 \times 10^{-11}$ | 0.51 |
| 7p13 | tCamp | rs217385 | 24 (22) | *NPC1L1* (21) | T/G | 0.43 | 0 | −0.041 | $6.3 \times 10^{-15}$ | 0.62 |
| 9q34.2 | tCamp | rs2519093 | 38 (23) | *ABO* (0) | T/C | 0.22 | 0 | 0.045 | $1.6 \times 10^{-12}$ | 0.51 |
| 10q25.3 | tSito | rs2286779 | 4 (4) | *PNLIPRP2* (0) | C/G | 0.53 | 15 | 0.054 | $1.9 \times 10^{-15}$ | 0.64 |
| 12q24.31 | tSito | rs10846744 | 5 (5) | *SCARB1* (0) | C/G | 0.17 | 68 | 0.063 | $2.9 \times 10^{-12}$ | 0.50 |
| 19q13.32 | eStig | rs7412 | 1 (1) | *APOE* (0) | T/C | 0.088 | 0 | −0.073 | $1.9 \times 10^{-14}$ | 0.83 |

For the locus 2p21, four independent variants were discovered by Cojo-Select analysis. For the other loci, only a single independent variant was found. For each variant, we present cytoband, best associated trait at this locus, rsID and corresponding statistics (of fixed effect meta-analysis of gene-dose effects). For the 2p21 variants, conditional effect estimates and p-values are shown. The sizes of the 99%, respectively 95% credible sets are also provided. Corresponding variants are annotated in Supplementary Data S8. Annotations are in accordance with genome-build GRCh37. Trait abbreviations are explained in Supplementary Data S18.

In total, the four independent SNPs of this locus explain 6% of total sitosterol variance.

*9q34.2.* We also confirm the second locus discovered in our former GWAS, 9q34.2 (*ABO*). Total campesterol is the best associated trait here. The top-associated variant is rs2519093 ($p = 1.6 \times 10^{-12}$) which is in perfect LD in terms of Lewontin's D'[31] with our formerly described variant rs657152 (D' = 1, $r^2$ = 0.34). In our former work, rs657152 showed strong LD with rs8176719 coding for the blood group O. Accordingly, a recessive model for rs657152 could be assumed. Analysing the haplotype frequencies of the T/C alleles at rs2519093 and the C/− alleles of rs8176719, it revealed that the T allele of rs2519093 associated with higher campesterol implies the C allele of rs8176719 corresponding to non-O blood groups (see Supplementary Fig. S7). Thus, these results are in agreement

with our former finding that non-O blood groups are associated with higher phytosterols[9].

Indeed, a more detailed analysis of total campesterol associations revealed that a recessive model of inheritance can be assumed for both, allele C of rs2519093 and allele "−" of rs8176719 ($p = 4.6 \times 10^{-4}$, respectively $p = 1.7 \times 10^{-4}$ for testing an additional heterozygote effect under an additive model). Accordingly, the recessive model of both SNPs showed stronger effect sizes and significances compared to the additive model ($\beta = −0.059$, $p = 4.0 \times 10^{-13}$, respectively $\beta = −0.054$, $p = 7.8 \times 10^{-11}$, see Supplementary Data S10).

The locus is co-associated with cholesterol, CAD and an eQTL of *ABO* in blood (see Supplementary Fig. S6). Again, the association with total cholesterol is considerably weaker as with total campesterol (explained variance for total campesterol 0.5%,

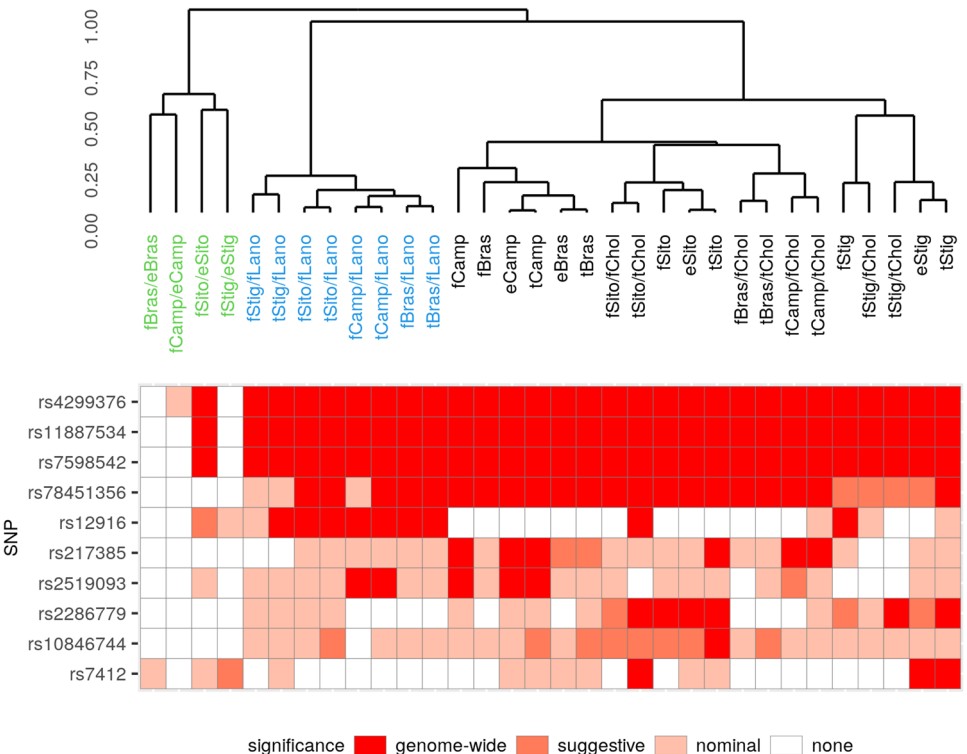

**Fig. 3 Association results per top-SNP.** Overview of the ten independent SNPs found in our genome-wide association analysis and their respective trait associations. Colour coding corresponds to level of significance of the respective traits. The distance function used for the dendrogram corresponds to the partial correlations of the analysed phytosterol traits (see 'Methods' for details). Ratios of phytosterols to lanosterol (blue) and free to esterified ratios (green) cluster separately. The pattern of significances mirror this correlation structure. The hit at 5q13.3 (rs12916, fifth row) is driven by a strong lanosterol association. A locus-wise presentation of associations can be found in Supplementary Data S7.

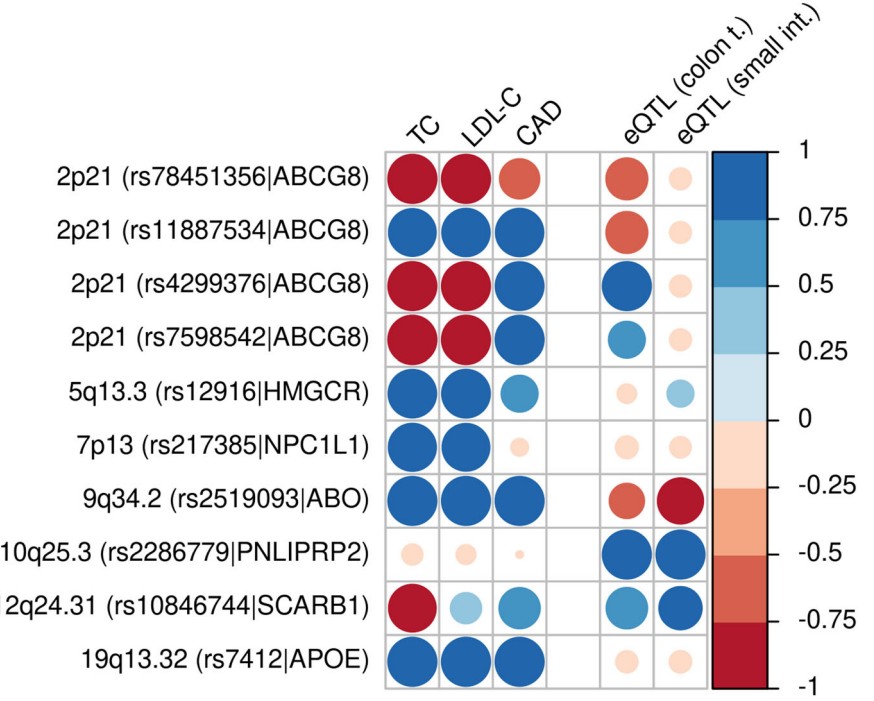

**Fig. 4 Results of colocalization analysis.** The ten independent meta-GWAS signals were subjected to colocalization analysis with other traits and eQTLs in colon tissue and small intestine. We depict the posterior probabilities of $H_4$ (evidence of colocalization) in blue and the negative posterior probabilities of $H_3$ (no evidence of colocalization) in red. Darkness of colour and size of circles correspond to the numerical value of the corresponding posterior probability. EQTLs are restricted to interesting findings. Analysis of eQTLs in other tissues are shown in Supplementary Fig. S6. Numerical results are shown in Supplementary Data S9. TC = Total cholesterol, LDL-C = low-dense lipoprotein-cholesterol, CAD = coronary artery disease, eQTL = expression quantitative trait locus.

for total cholesterol 0.1%). The 99% credible set contains 38 variants.

**Novel loci**. We summarize the results of the five novel loci in the following, ordered by position:

*5q13.3.* At 5q13.3, the top hit rs12916 is associated with quotients of phytosterols and free lanosterol (best associated trait total brassicasterol to lanosterol, $p = 2.3 \times 10^{-11}$). Total phytosterols alone are not associated. The locus is a known cholesterol locus with *HMGCR* as the causal gene. We, therefore, consider this association driven by zoosterols rather than phytosterol. Accordingly, the locus is colocalized with cholesterol (PP4 = 97.9%), and weakly, with CAD (PP4 = 56%).

*7p13.* Strongest association at this locus was observed for rs217385 with total campesterol ($p = 6.3 \times 10^{-15}$). The ratio of campesterol and cholesterol is also significantly associated as well as other campesterol traits and total sitosterol. Other variants of the locus are in LD with SNPs associated with cholesterol[32]. The locus colocalizes with cholesterol (PP4 = 95.9%) but this signal is clearly weaker explaining much lesser variance than for total campesterol (0.6% for total campesterol compared to 0.05% for total cholesterol). The 99% credible set contains 24 variants. The most plausible candidate is *NPC1L1* which transports several sterols from intestine to enterocytes[33]. In line with this, pharmaceutical inactivation of *NPC1L1* by ezetimibe is an established treatment against sitosterolemia[34].

*10q25.3.* This locus is driven by a total sitosterol association ($p = 1.9 \times 10^{-15}$). Other sitosterol traits including normalized total sitosterol as well as total stigmasterol are also associated with genome-wide significance. The top-variant is in some LD with variants reported to be associated with phospholipids (SNP rs10885997, $r^2 = 0.64$[35]). No colocalizations with cholesterol or CAD were observed. The 99% credible set contains four variants, all with similar posterior probability due to perfect LD. Among them, rs4751995 showed the highest CADD score of 10.4. This SNP is a variant of *PNLIPRP2* appearing as both, an intron and an exon variant depending on splicing according to HG38 Genome built. The SNP is also a strong cis-eQTL of this gene in several tissues including colon, pancreas, stomach and small intestine. Accordingly, eQTL colocalizations in these tissues were observed. The gene is also biologically plausible because PNLIPRP2 shows high hydrolytic activity on phospholipid bile salt micelles[36]. Bile salt micelles influence phytosterol levels due to different affinities to zoo- and phytosterols[37].

*12q24.31.* The strongest association at this locus was observed for rs10846744 with total sitosterol. Other associated traits comprise esterified sitosterol and the ratios free sitosterol to free cholesterol and total sitosterol to lanosterol. A week evidence for colocalization with CAD (PP4 = 70%) was found, but interestingly, not with total cholesterol (PP3 = 94%) or HDL-C (PP3 = 100%) despite of the fact that the locus was described for associations with different lipid traits[38,39]. Our lead variant is also not in LD ($r^2 = 0.019$) with rs838880 reported in Teslovich et al.[38]. Accordingly, the SNP explains considerably more variance of sitosterol as compared to total cholesterol or LDL-C (0.5% vs. 0.013% respectively 0.02%). Regarding eQTLs the locus (weakly) colocalizes with an eQTL of *SCARB1* in small intestine and colon tissue[40] (PP4 = 77%, respectively PP4 = 71%). According to the GWAS catalogue, the locus is also associated with PLA2 activity and mass. The 99% credible set comprises five variants, all in LD with the top-variant. These SNPs are intronic variants of

*SCARB1* with no relevant deleteriousness prediction (maximum CADD score 6.0). The scavenger receptor class B type I (SCARB1 or SR-BI) is a receptor of HDL and facilitates cholesterol delivery to steroidogenic tissues and cholesterol excretion in the liver[41,42]. As a possible mechanistic explanation of our results, we suppose that increased expression of *SCARB1* improves uptake of cholesterol from micelles by enterocytes. In response, an increased phytosterol uptake by micelles is conceivable. This is in line with the SNP's unidirectional effects on sitosterol and *SCARB1* gene-expression.

*19q13.32.* Finally, we detected a genome-wide significant association with esterified and total stigmasterol at 19q13.32. The lead-SNP was rs7412 and the 99% credible set contains only this SNP. The SNP is a known miss-sense mutation of *APOE* (R176C), representing the *APOE*-E2 allele. The locus colocalizes with cholesterol (PP4 = 100%) and CAD (PP4 = 99.7%) but no eQTLs. The cholesterol effect of this locus is larger than that of stigmasterol (1.9% explained variance for total cholesterol, 3.8% for LDL-C compared to 0.8% for the best associated trait esterified stigmasterol). Therefore, we consider this locus as driven by zoosterol rather than phytosterol associations. The effect directions of the variant on cholesterol and stigmasterol are identical at this locus. This is in agreement with the observation that phytosterols are accumulated in *APOE* knock-out mice but not in *LDLR* knock-out mice[43], which was explained by increased blockage of sterol excretion rather than by absorption, which would expected to be lower in case of increased cholesterol synthesis[44].

**Heritability**. To assess the potential for future genome-wide association studies of phytosterol traits, we estimated their chip-heritability. Estimates were significant throughout and effect sizes are moderate to large. The largest heritability was estimated for total campesterol to cholesterol ratio and esterified campesterol ($h^2 = 72\%$, $p < 1.3 \times 10^{-5}$). The heritability estimate of total sitosterol showing strongest associations in our study was 63%. Since the seven independent variants found for this trait explain 7.4% of the variance, further variants for this trait are likely to exist.

Interestingly, quotients of free to esterified phytosterols and quotients of phytosterols to lanosterol yielded relatively small heritability estimates, which is in agreement with the fact that none of our variants are detected on the basis of these traits, except for the *HMGCR* locus which is driven by a strong lanosterol association. All heritability results can be found at Supplementary Data S11.

**Look-up of lipid loci**. We performed a look-up of 1,600 independent lipid loci reported in literature. Among those, 220 showed nominal significance with at least one of our phytosterol traits ($p_{min} < 3.48 \times 10^{-3}$ corresponding to a significance threshold of 5% accounting for multiple trait testing, see methods). This constitutes a strong enrichment of OR = 2.65 ($p = 7.8 \times 10^{-41}$). Five loci where detected with suggestive significance ($p < 1.0 \times 10^{-6}$), namely 11q23.3 (*TAGLN*, *PCSK7*, $p = 5.1 \times 10^{-8}$ for free sitosterol), 20q13.12 (*HNF4A*, $p = 7.0 \times 10^{-8}$ for total stigmasterol to cholesterol ratio), 2p24.1 (*APOB*, $p = 1.2 \times 10^{-7}$ for free to esterified sitosterol ratio), 5q13.3 (*ANKDD1B*, $p = 1.6 \times 10^{-7}$ for total brassicasterol to lanosterol ratio), 9p22.3 (*TTC39B*, $p = 5.6 \times 10^{-7}$ for free campesterol to cholesterol ratio). These loci could be considered further candidates requiring replication. Full look-up results can be found in Supplementary Data S12.

**Table 2 Mendelian randomization results.**

| Parameter | X | Y | Causal estimate | se | *p*-value | # SNPs |
|---|---|---|---|---|---|---|
| α | SIT | TC | 0.419 | 0.052 | $7.6 \times 10^{-16}$ | 6 |
| β | TC | CAD | 0.347 | 0.059 | $4.8 \times 10^{-9}$ | 36 |
| γ | SIT | CAD | 0.308 | 0.065 | $1.9 \times 10^{-6}$ | 6 |
| Indirect effect (α * β) mediated by TC | SIT | CAD | 0.145 | 0.031 | $2.2 \times 10^{-6}$ | |
| Direct effect (γ − α * β) | SIT | CAD | 0.163 | 0.072 | $2.3 \times 10^{-2}$ | |

We performed Mendelian randomization analyses of total sitosterol, total cholesterol and CAD. A schematic figure of the investigated causal relationships is displayed in Supplementary Fig. S8. We provide single causal effect estimates based on the given number of SNPs used as instruments (method: inverse-variance weighting). From these estimates, the direct effect of total sitosterol on CAD and the indirect effect mediated by total cholesterol are calculated as described in the 'Methods' section. Both causal effects were positive and significant while the direct effect is slightly larger than the indirect effect. Considering more restricted sets of instrumental variables provided similar results (see Supplementary Data S14) as well as applying different methods of Mendelian randomization analysis (Supplementary Fig. S9).

Since no specific signals were found for the free to esterified phytosterol ratios, we looked up variants in the genes *LCAT, ACAT, SOAT1* and *SOAT2* involved in phytosterol esterification. It revealed that no suggestive hits were present (see Supplementary Data S13).

**Mendelian randomization analysis.** Mendelian randomization analyses were performed for total sitosterol, total cholesterol and CAD in Europeans first (see Supplementary Fig. S8). Using six independent variants of total sitosterol identified in the present study, causal positive effects could be found for total sitosterol on cholesterol and for total sitosterol on CAD. The effect of total cholesterol on CAD was also positive and significant (see Table 2). Based on these results, we determined the direct effect of total sitosterol on CAD and the indirect effect mediated via cholesterol. It turned out that both are positive and significant. The direct effect constitutes 53% of the total effect, i.e. is roughly in the same order as the indirect effect.

This result was confirmed by analysing normalized sitosterol also showing a positive causal effect on CAD (Effect: 0.32, $p = 2.7 \times 10^{-6}$, Supplementary Data S14, see Supplementary Data S16 and S17 for single SNP statistics). Sensitivity analyses considering only variants of 2p21 as instruments or restricting to strong instruments of total cholesterol did not change the results (see Supplementary Data S14). Sensitivity analysis applying other MR methods showed consistent effects throughout (see Supplementary Fig. S9).

Finally, we repeated the analysis considering CAD summary statistics from Japanese subjects. We observed essentially the same results (see Supplementary Data S15, Supplementary Fig. S10).

## Discussion

In this genome-wide meta-analysis of phytosterol traits also considering free and esterified traits we significantly increased the sample size of our previously published single study GWAS (up to 9758 compared to 1495 of our previous study). We systematically compared genetic effects on absolute phytosterols, phytosterol to zoosterol ratios and esterification representing different facets of phytosterol metabolism including markers of resorption and synthesis of cholesterol. We identified ten independent genome-wide significant associations at seven loci, comprising five new loci robustly associated with multiple traits and related to functionally plausible genes. Since our associations provide strong genetic instruments, we performed a comprehensive Mendelian randomization analysis of the causal relationships of sitosterol, total cholesterol and CAD. It revealed a causal effect of higher plasma sitosterol levels on increased CAD risk that is only partly mediated by the increase of total cholesterol levels, thus supporting an atherogenic effect of phytosterols.

In our previous single study meta-analysis[9], we detected three independent variants of phytosterol traits. Two of them were located at 2p21 (*ABCG5/8*), while another one was located at 9q34.2 (*ABO*). We could confirm these findings in our present meta-analysis comprising a considerably larger sample size. We could also confirm that the genetic model at 9q34.2 could be assumed to be recessive, i.e. carriers of non-O blood groups show higher phytosterol levels, which is consistent with higher cholesterol levels[45] and higher CAD risk[46] of non-O carriers. The *ABO* locus is notorious for its pleiotropic effects on other traits including E-Selectin and other lipid species[47,48].

However, with respect to the 2p21 locus, our fine-mapping analysis with increased sample size revealed four rather than the previously reported two independent associations with putatively different functional mechanisms. While rs4299376 is a strong eQTL of *ABCG8* in colon tissue, rs11887534 acts via a non-synonymous coding mutation. The situation for the third SNP, rs7598542, is less clear because on one hand it colocalizes weakly with an eQTL of *ABCG8* in colon tissue, but on the other hand, the credible set also contains non-synonymous coding mutations. The fourth variant rs78451356 is outside of the haplo-block of the three other variants and the respective credible set contains functional intron variants of *ABCG5*. Of note, the independent variants at this locus do not show any colocalization signals with total cholesterol except for rs11887534 for which, however, the explained variance of sitosterol was much higher than that for total cholesterol. Thus, we conclude that this is a primary phytosterol locus and that observed associations with cholesterol are secondary to that.

Among the five new loci, the 5q13.3 (*HMGCR*) locus associated with total brassicasterol to lanosterol was likely driven by lanosterol association. Likewise, the 19q13.32 (*APOE*) associated with esterified stigmasterol might also be driven by associations with other lipid species because the locus colocalizes with a total cholesterol association explaining a larger amount of total variance. It is worthwhile to mention that normalization to cholesterol or lanosterol, respectively, can induce genetic associations driven by these traits. Therefore, we recommend considering both, raw and normalized traits.

For the other three loci comprising 7p13, 10q25.3, 12q24.31 functionally plausible genes could be assigned, namely *NPC1L1, PNLIPRP2* and *SCARB*1. *NPC1L1* inactivation by ezetimibe already showed sitosterol lowering effects[33,34]. Nissinen et al. also found (small) effects of *NPC1L1* variants on phytosterols in children[49]. *PNLIPRP2* and *SCARB1* both interact with micelles, which in turn express competitive zoo- and phytosterol uptake. There is experimental evidence regarding involvement of SCARB1 in sterol uptake shown by cell-culture experiments[50,51] and over-expression in mice[52,53]. In contrast, such an effect could not be observed in *SCARB1* knock-out mice[54]. Of note, 10q25.3 showed no colocalization with total cholesterol and 7p13 and 12q24.31 showed colocalization explaining much less total variance of cholesterol compared to the associated phytosterol traits. Thus, we again conclude that these loci are primary

phytosterol loci and that cholesterol associations are down-stream effects.

A limitation of our association analyses is that we adjusted for the binary variable "lipid lowering medication" as determined by ATC category "C10". This does not consider dosing schemes, which are scarcely available in population-based studies. Moreover, we did not distinguish between sub-categories of ATC C10. In the majority of cases, statins were prescribed (e.g. LIFE-Adult: 95%, LIFE-Heart: 97% of those receiving a drug from the C10 category). All other categories were much less prescribed (6% respectively 5%). Ezetimibe was rarely prescribed (<2% of cases).

In our analysis, phytosterols showed a moderate to high heritability and our estimates are in the same order of magnitude as those of twin studies[7]. It needs to be pointed out that our estimates refer to the so-called "chip heritability", i.e. variants which are covered by the chosen genotyping platform including well-imputable variants. Thus, these estimates are a lower bound of the total heritability. In our study, we estimated for example a heritability of 63% for sitosterol. On the other hand, discovered variants only explained 7.4% of variance. This suggests that phytosterols are complex traits and that there are more variants to be discovered in future meta-GWAS efforts. Accordingly, our look-up of loci associated with other lipid traits (total cholesterol, LDL-C, HDL-C, triglycerides) revealed several additional associations with nominal or suggestive significance. Larger sample sizes are required to validate these associations and to find low frequency variants. Moreover, studies in other than European ancestries are required to reveal any ethnicity-specific variants or effects.

The close inter-relationship of phyto- and zoosterols raises questions regarding causes and consequences of observed genetic associations and with respect to the conversely discussed relationship of phytosterols and CAD. For a formal analysis of the causal inter-relationships between total sitosterol, total cholesterol and CAD, we applied Mendelian randomization. We aimed to distinguish between a direct causal effects of total sitosterol on CAD and an indirect effect mediated by total cholesterol. This is not trivial because it requires independent genetic instruments for total sitosterol and total cholesterol while genetic associations are often observed for both traits in parallel. We therefore selected genetic instruments for which type I pleiotropy can be excluded as far as possible, either based on the functional role of the candidate gene or by sole or particularly strong genetic associations for one of the traits only. For sitosterol instruments, we considered six independent genome-wide significant variants discovered in the present study. All showed clearly stronger effect sizes with sitosterol than with total cholesterol. For total cholesterol, we considered 36 variants[55] excluding all cytobands with phytosterol associations. In sensitivity analyses, we also restricted instruments of sitosterol to the independent variants of the 2p21 locus for which a clear functional role in phytosterol excretion is established. Instruments of total cholesterol were also restricted to the 14 strongest associations. We further considered methods for MR, which are more robust regarding type I pleiotropies. Similar results were observed throughout suggesting that our MR analysis is not biased by type I pleiotropies. Interestingly, we could show both, a significant direct effect of increased total sitosterol on elevated CAD risk and a significant indirect effect mediated by total cholesterol. Effects were in roughly the same order of magnitude. This observation was confirmed by considering normalized sitosterol, again showing a positive causal effect on CAD risk. Considering CAD summary statistics from Japanese samples yielded roughly the same results. It needs to be pointed out, however, that this type of Mendelian randomization analysis is performed under the assumption of comparable instrumental variable effects on sterols between Europeans and Japanese.

Comparing allelic frequencies of instrumental variants between these ethnicities revealed larger differences. Thus, further investigations are required to validate our causal estimates for non-European ethnicities. As another limitation, by Mendelian randomization one estimates the effect of a small live-long increase of total serum sitosterol on total cholesterol or CAD risk. The effects of short-term dietary or pharmaceutical interventions cannot be estimated by this method.

In summary, our study extends the number of variants and loci associated with serum phytosterol traits. It also provides further candidate genes to be confirmed in future studies. Contributing to the ongoing discussion of a potential role of phytosterols on the risk of CAD, our Mendelian randomization analyses provided evidence for both, a direct and a cholesterol-mediated detrimental effect of sitosterol on CAD risk.

## Methods
The overall analysis workflow and study design is depicted in Supplementary Fig. S11.

**Contributing studies**. Six studies contributed to the present analysis: KORA[56,57], LIFE-Adult[58], LIFE-Heart[59], LURIC[60], the Sorbs[61,62] and the Young Finns study[63] (YFS). All study participants were of European ancestry. Brief study descriptions are provided in Supplementary Data S1. Reported study and phenotype characteristics, as well as technical details, are presented in Supplementary Data S2 for each study. Information on genotyping, phenotyping, quality control and data analysis are summarised below.

**Measurement of sterols**. The following parameters were measured in some or all of the contributing studies: serum concentrations of free and esterified brassicasterol, campesterol, sitosterol, stigmasterol, cholesterol and free lanosterol. In KORA, LIFE-Adult, LIFE-Heart and Sorbs sterol measurements were performed centrally at the Institute of Laboratory Medicine, University of Leipzig using liquid chromatography tandem mass spectrometry following the same analytical protocol for all studies. The measurement technique is explained in detail in[64]. We performed adjustment of measured quantities regarding batch-effects by treating time of measurement as batch parameter. Function ComBat of the R-package "sva" was applied for that purpose[65] (R Core Team. R: A Language and Environment for Statistical Computing.Vienna, Austria. https://www.R-project.org/). See Supplementary Data S19 for a complete overview of R-package versions used in the present work.

In LURIC, serum levels of total brassicasterol, campesterol, lanosterol, sitosterol, stigmasterol as well as free and total cholesterol were measured with gas-chromatography mass-spectrometry. In YFS, serum levels of total campesterol, cholesterol and sitosterol were also determined using gas-chromatography mass spectrometry.

Descriptive statistics of available traits per study can be found at Supplementary Data S2.

**Trait definition and hierarchical clustering**. Genome-wide association analyses were performed for the following traits if available: Total, free and esterified phytosterols (12 traits at maximum). Moreover, we considered a number of physiologically relevant ratios, namely those of free to esterified phytosterols (4 traits), free and total phytosterols to lanosterol (8 traits), and to total cholesterol (8 traits), respectively. Ratios represent reaction equilibria of phytosterol esterification and phytosterols normalized to lanosterol or to cholesterol as measures of endogenous cholesterol synthesis, respectively cholesterol absorption. Thus, at maximum, 32 traits were analysed.

To visualize correlation structure between our traits, we performed hierarchical clustering. This analysis is based on the phytosterol data of the studies KORA, LIFE-Adult, LIFE-Heart and Sorbs measured by the same method. For the clustering, we consider partial correlation coefficients as measures of similarity of traits controlling for the covariates age, sex, log(BMI), diabetes status, lipid lowering medication and study. Traits were log-transformed prior to analysis. Clustering was performed using the "hclust" package of the software R.

**Genotyping and imputation**. Genotypes were measured by using SNP microarrays: Affymetrix Gene Chip Human Mapping 500 K Array Set (Sorbs), Affymetrix Axiom CEU1 (LIFE-Adult, LIFE-Heart), Affymetrix custom array (LIFE-Heart), Affymetrix Genome-Wide Human SNP Array 6.0 (LURIC, Sorbs), Illumina 200k MetaboChip (LURIC), Illumina Omni 2.5/Illumina Omni Express (KORA) and Illumina Human 670k BeadChip (YFS). Sample and SNP quality control was performed according to study specific criteria, see Supplementary Data S2 for details.

Genotypes were imputed using IMPUTE2[66] based on 1000 Genomes Phase 1 reference panel (LIFE-Adult, LIFE-Heart, LURIC, Sorbs, YFS) or 1000 Genomes Phase 3 reference panel (KORA). Genotype data was translated to forward strand annotation using NCBI b37 (hg19) coordinates.

**Single study genome-wide association analysis.** A standardized analysis plan was developed to harmonize genome-wide association analyses of single studies. Analyses were performed centrally for the cohorts KORA, LIFE-Adult, LIFE-Heart and Sorbs. Analysts of the two external cohorts, LURIC and YFS, were asked to follow the same analysis plan.

Traits were log-transformed to approximate Gaussian distributions. To minimise confounding, traits were adjusted for age, sex, log(BMI), diabetes status and lipid lowering medication (Anatomical Therapeutic Chemical (ATC) code "C10"). Regression analyses were also adjusted for genetic principal components when indicated. Due to excess relatedness in the Sorbs study[67], respective traits were also adjusted for the relatedness structure applying mixed model analysis as represented in the "polygenic" function of the "GenABEL" package of R.

Association analyses were performed using PLINK 1.9[68] (LIFE-Adult, LIFE-Heart, Sorbs), PLINK 2.0 (LURIC, KORA) or SNPTEST 2.5[69] (YFS) assuming an additive gene-dose model. X-chromosomal markers were analysed assuming total X-inactivation, i.e. male genotypes were coded as A = 0 and B = 2 and female genotypes are coded as AA = 0, AB = 1 and BB = 2. As effect estimates, slopes of the gene-dose in linear regression analysis and respective standard errors are reported. P-values correspond to two-sided testing.

Sample sizes and SNP numbers available per study and trait are provided in S3.

**Quality control of single study association results.** Summary statistics of all studies were checked and harmonized using EasyQC[70]. SNPs not in the reference panel (1000 Genomes phase 1, version 3, European ancestry), with missing values in alleles (effect allele, effect allele frequency) or statistics (e.g. beta estimates, imputation quality score), mismatching alleles or mismatching chromosomal position with respect to the reference were discarded. SNPs were filtered for weighted minor allele frequency (MAF) > 2% which corresponds to minor allele count >17 as calculated for the smallest study (YFS, $N = 432$). Genotyped SNPs were filtered for call rate >95% and p(HWE) > $10^{-6}$. Imputed SNPs were filtered for imputation quality score >0.5 and for deviation from reference allele frequency <20%. Finally, the alleles were harmonized so that the same effect allele was used in all studies. Number of quality controlled SNPs per study is presented in Supplementary Data S3.

Variance inflation factor λ was calculated for single study GWAS. Test statistics were corrected by genomic control[71] if λ > 1.

**Meta-analysis.** Altogether 32 traits and up to 9758 samples per trait were meta-analysed. Fixed effects inverse variance meta-analysis of single study gene-dose effects were performed as primary statistics. Random effects meta-analysis results were also reported. Meta-analysis results were filtered for number of contributing studies >2 and heterogeneity $I^2 < 0.7$.

The number of resulting SNPs ranged from 7,827,943 to 8,212,880 in dependence on the trait analysed (see Supplementary Data S3 for details).

A p-value of <$5 \times 10^{-8}$ was considered genome-wide significant (two-sided test of fixed effect). We also visually inspected the regional association plots of genome-wide significant loci and removed those with lack of support, i.e. no other variant with at least suggestive significance ($p < 10^{-6}$).

Since for one of the loci (*ABO*) a recessive mode of inheritance can be assumed, we analysed possible deviations from the standard additive model in cohorts for which we had access to the raw genotype data (KORA, LIFE-Adult, LIFE-Heart, Sorbs) using the "DOMDEV" option of Plink. The null-hypothesis of this test is that heterozygote effects are zero under an additive model.

**Hit annotation.** Annotation of Meta-analysis results was done by an in-house workflow (see ref. [72] for details). In brief, linkage disequilibrium (LD) between markers was calculated based on genetic data from 1000 Genomes Phase 1, version 3 reference panel for European samples. Priority pruning of the top-list was performed by assuming a variant as tagged when the variant is in LD ($r^2 \geq 0.5$) with a tag-SNP of stronger association with any trait. Loci are defined by cytobands.

Lead SNPs of loci are defined as the tag SNPs showing the strongest association with any trait. All genes within 50 kilobases (kb) distance and up to four genes within a 250 kb distance to a SNP according to Ensembl[73] are reported as candidate genes due to proximity. SNPs were annotated by further resources comprising known trait associations via LD-based lookup ($r^2 \geq 0.3$) of the most recent GWAS Catalog[74], expression quantitative trait loci (eQTLs) by LD-based lookup ($r^2 \geq 0.3$) based on Genotype-Tissue Expression (GTEx v7)[75] and (updated) own data[76] and by various deleteriousness scores including CADD[77] and RegulomeDB[78]. All resources were downloaded at July 1st, 2020.

We also annotated the nearby genes and eQTL genes per locus by assigning respective pathways retrieved from KEGG, GO, DOSE[79], and reactome (downloaded April 15th, 2020).

**Conditional and joint analyses, explained variances.** To identify secondary hits per locus, we considered the best-associated trait and applied the tool GCTA (version 1.92.0beta3)[80]. First, we performed stepwise model selection (cojo-slct) to identify independent variants per locus. If more than one such signal was observed, we calculated conditional statistics (cojo-cond)[81] for every independent variant by controlling for the other independent variants, respectively. As LD reference panel we used the combination of available genotypes of LIFE-Adult and LIFE-Heart ($n = 13,369$).

After determining independent SNPs via Cojo analysis, we calculated their respective explained variance using the formula $r^2 = \beta^2/(\beta^2 + N \cdot se(\beta)^2)$[82], where β is the fixed meta-effect of Beta estimates of the single study linear regression analyses, $se(\beta)$ its standard error and N the total sample size. For the 2p21 locus, the conditional statistics were used. Total explained variance is calculated by summing up the explained variances of single independent SNPs contributing to the respective trait.

**Credible set analyses.** After determination of independent signals, we aimed at identifying the respective set of SNPs containing the causal variant with high certainty. For this purpose, we considered the set of SNPs within ±500 kb of the independent lead SNP and their respective effect estimates and standard errors[83,84]. In case of more than one independent variant per locus, conditional statistics were used per independent variant. We then calculated respective Approximate Bayes Factors (ABF) by applying the R-package "gtx". The required prior distribution of the standard deviation was constructed empirically by the difference of the 97.5th and the 2.5th percentile of SNP effects of the respective locus divided by (2*1.96). In our data, this quantity ranged in between 0.014 (locus 9q34.2) and 0.051 (2p21).

Derived ABFs were used to calculate the posterior probability of a variant being causal for the observed association. We ordered variants in descending order of their posterior probability and determined the respective cumulative probability. Applying a cut-off of 99% cumulative probability yielded the 99% credible set of SNPs for the respective variant. We also considered the relaxed cut-off of 95%. Variants of the credible sets were annotated as described above. The CADD score was considered as primary criterion to identify functionally relevant variants within the credible sets.

**Heritability and look-up of lipid candidate loci.** We estimated the heritability of all considered GWAS traits using the raw genotype data of LIFE-Adult and LIFE-Heart and applying GCTA. This approach results in the so called "chip-heritability"[85].

We also systematically searched for co-associations of our traits with reported lipid loci to detect possible associations, which did not achieve genome-wide significance in our study. Lipid loci were retrieved from the GWAS Catalog[86] by searching for total cholesterol (TC, trait ID in the experimental factor ontology: EFO_0004574), low density lipoprotein-cholesterol (LDL-C, EFO_0004611), high density lipoprotein-cholesterol (HDL-C, EFO_0004612) and triglycerides (TG, EFO_0004530). Download was performed at 20th August 2020. We only considered variants for which genome-wide significance was reported (3770 unique SNPs). Of those, 3067 were available in our data and high quality association results were available for 3003 of them. After pruning, 1600 independent variants were obtained.

Phytosterol traits are considered co-associated if achieving a minimum p-value <$3.48 \times 10^{-3}$ across all analysed traits. This corresponds to a 5% significance threshold accounting for multiple trait testing and was obtained on the basis of the empirical p-value distribution of our 8,299,000 SNPs with association results. We tested for an enrichment of co-associated phytosterol traits using a one-sided exact binomial test.

Finally, we searched for candidate genes of phytosterol esterification, namely *LCAT*, *ACAT*, *SOAT1* and *SOAT2* by considering all SNPs within a 500 kb range around these genes.

**Colocalization analyses.** We tested whether the independent loci coincide with loci of other lipid traits, coronary artery disease (CAD) or cis-eQTLs of candidate genes in different tissues. The latter could provide a potential functional explanation of the considered variant e.g. for those for which no causal non-synonymous coding mutation could be detected in the respective credible set. For 2p21, conditional statistics were used for that purpose. Publicly available summary statistics of the considered traits are available from recent GWAS[55,87]. Cis-eQTLs were retrieved from GTEx v7 (whole blood, esophagus mucosa, small intestine, colon transvers, colon sigmoid, adrenal gland, liver, pancreas)[29]. Coincidence of signals was tested by pairwise colocalization analyses of loci[88]. This method evaluates the posterior probability of five hypotheses (H₀: no associations within locus; H₁,₂: associations with either trait 1 or trait 2 only, H₃: association with both traits but different SNPs, H₄: association with both traits with the same SNP—evidence for colocalization). Posterior probabilities of these five hypotheses are defined as positive and sum up to 100%. We consider a posterior probability of ≥75% as sufficient to support one of the hypotheses. Loci were again defined by a ± 500 kb window around the respective lead SNPs.

**Mendelian randomization**. The role of phytosterols in the development of coronary artery disease is controversially discussed. Therefore, we exploited the results of our genome-wide association analysis to perform Mendelian randomization analyses. We aimed at answering the question whether there is a causal relationship of phytosterols on CAD and to what extend this effect is mediated by cholesterol.

Since the strongest instrumental variables were obtained for total sitosterol, we focused on this trait throughout. Six independent genome-wide associations identified in our meta-GWAS were considered as instruments, namely three independent variants from the 2p21 locus and the three genome-wide significant hits at 7p13, 10q25.3 and 12q24.31, respectively. The fourth independent variant from 2p21 could not be used due to missing CAD summary statistics. To avoid any biases due to possible type I pleiotropies (e.g. SNPs directly influencing multiple traits in parallel[89]), we also performed a sensitivity analysis restricting to the independent SNPs of the 2p21 locus only, since this locus is functionally well established for its role in phytosterol excretion.

For total cholesterol, we used 36 SNPs as instrumental variables not associated with phytosterol levels (summary statistics from Surakka et al.[55]). This was achieved by removing cytobands with phytosterol associations. Again, to avoid type I pleiotropy, we also performed these analyses restricting to strong instruments, i.e. variants with $p < 10^{-20}$. Summary statistics for CAD were retrieved from van der Harst et al.[90].

For validation purposes, we also estimated the causal effect of the ratio of total sitosterol to cholesterol on CAD. The same variants were considered as for total sitosterol. However, since larger heterogeneity was observed for the total sitosterol to cholesterol trait, we removed YFS to calculate the instrumental effects.

Finally, to analyse possible translations to other ethnicities, we performed Mendelian randomization analyses using CAD summary statistics from Japanese subjects[91] but assuming the same instrumental effects for total phytosterol and cholesterol as observed in Europeans. Here, we considered seven instruments for total phytosterol and 26 instruments for total cholesterol. All analyses are restricted to subsets of instruments for which all required genetic summary statistics (i.e. for total sitosterol, total cholesterol and CAD) are available.

The direct and indirect causal effect of total sitosterol is estimated as recommended by Burgess et al.[92], i.e. by the following three steps (see also Supplementary Fig. S8):

(1) Estimate the total causal effect of total sitosterol on CAD by standard MR analysis ($\gamma$)
(2) Estimate the causal effect of total sitosterol on total cholesterol ($\alpha$)
(3) Estimate the causal effect of total cholesterol on CAD ($\beta$)

Then, the indirect effect constitutes on the product of $\alpha$ and $\beta$, while the direct effect can be estimated by $\gamma$ minus the indirect effect. For causal effect estimation, we used the inverse-variance weighting method as implemented in the R package "MendelianRandomization". Other methods (MR-Egger, Simple median and weighted median) were also considered for sensitivity analysis.

**Reporting summary**. Further information on research design is available in the Nature Research Reporting Summary linked to this article.

## Data availability

Genome-wide summary statistics generated in this study have been deposited at https://doi.org/10.5281/zenodo.5607612. Used public data bases are: Deleteriousness scores (http://www.regulomedb.org/), GWAS catalogue (https://www.ebi.ac.uk/gwas/api/search/downloads/full), eQTLs (ftp://ftp.ncbi.nlm.nih.gov/eqtl/original_submissions/FHS_eQTL/). DOSE and Reactome pathways were retrieved via respective R-packages (see Supplementary Data S19). Genome-wide summary statistics of other studies were retrieved from web resources mentioned in the respective publications (see 'Methods').

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

## Acknowledgements

We gratefully acknowledge the contributions of P. Lichtner, G. Eckstein, G. Fischer, T. Strom and all other members of the Helmholtz Centre Munich genotyping staff in generating the SNP dataset as well as the contribution of all members of field staffs who were involved in the planning and conduct of the MONICA/KORA Augsburg studies. The KORA group consists of H.E. Wichmann (speaker), A. Peters, C. Meisinger, T. Illig, R. Holle, J. John and their co-

workers who are responsible for the design and conduct of the KORA studies. We thank Sylvia Henger for data quality control of LIFE-Adult and LIFE-Heart, Kay Olischer and Annegret Unger for technical assistance regarding LIFE-Heart, and Kerstin Wirkner for running the LIFE-Adult study center. Sincere thanks are given to Knut Krohn (Microarray Core Facility of the Interdisciplinary Centre for Clinical Research, University of Leipzig) for genotyping support of the Sorbs sample. We thank the LURIC study team who were either temporarily or permanently involved in patient recruitment as well as sample and data handling, in addition to the laboratory staff at the Ludwigshafen General Hospital and the Universities of Freiburg and Ulm, Germany. Finally, we express our appreciation to all participants of the contributing studies. The KORA research platform (KORA: Cooperative Research in the Region of Augsburg) and the MONICA Augsburg studies (Monitoring trends and determinants on cardiovascular diseases) were initiated and financed by the Helmholtz Zentrum München–National Research Center for Environmental Health, which is funded by the German Federal Ministry of Education, Science, Research and Technology and by the State of Bavaria. Part of this work was financed by the German National Genome Research Network (NGFN). Our research was supported within the Munich Center of Health Sciences (MC Health) as part of LMUinnovativ. LIFE-Heart and LIFE-Adult are funded by the Leipzig Research Center for Civilization Diseases (LIFE). LIFE is funded by means of the European Union, by the European Regional Development Fund (ERDF) and by means of the Free State of Saxony within the framework of the excellence initiative. The Sorbs study was supported by grants from the Collaborative Research Center funded by the German Research Foundation (CRC 1052; SPP 1629 TO 718/2), from the German Diabetes Association, from the DHFD (Diabetes Hilfs- und Forschungsfonds Deutschland) and from the German Center for Diabetes Research. LURIC was supported by the 7th Framework Program (integrated project AtheroRemo, grant agreement number 201668 and RiskyCAD, grant agreement number 305739) of the European Union. The Young Finns Study has been financially supported by the Academy of Finland: grants 322098, 286284, 134309 (Eye), 126925, 121584, 124282, 129378 (Salve), 117787 (Gendi), and 41071 (Skidi); the Social Insurance Institution of Finland; Competitive State Research Financing of the Expert Responsibility area of Kuopio, Tampere and Turku University Hospitals (grant X51001); Juho Vainio Foundation; Paavo Nurmi Foundation; Finnish Foundation for Cardiovascular Research; Finnish Cultural Foundation; The Sigrid Juselius Foundation; Tampere Tuberculosis Foundation; Emil Aaltonen Foundation; Yrjö Jahnsson Foundation; Signe and Ane Gyllenberg Foundation; Diabetes Research Foundation of Finnish Diabetes Association; EU Horizon 2020 (grant 755320 for TAXINOMISIS and grant 848146 for To Aition); European Research Council (grant 742927 for MULTIEPIGEN project); Tampere University Hospital Supporting Foundation and Finnish Society of Clinical Chemistry. Data analyses were supported by the German Federal Ministry of Education and Research (BMBF) within the framework of the e:Med research and funding concept (SYMPATH, grant # 01ZX1906B).

## Author contributions

Manuscript writing: M.S. Critical review of manuscript: K.H., J.P., H.K., R.B., B.I. and U.C. Design of the study: M.S., D.T. and U.C. Management of contributing studies: M.S., A.T, T.L., O.R., M.K., H.G., M.M.-N., M.S., M.L., W.M., A.P., D.T. and J.T. Genotyping: C.G., P.K. and T.M. Phytosterol measurements: U.C. Statistical analyses & Bioinformatics: M.S., K.H., J.P., A.G., M.E.K., G.E.D., P.P.M. and H.K. Interpretation of results: M.S., K.H., J.P., A.G., H.K. and U.C.

## Funding

## Competing interests

M. Scholz receives funding from Pfizer Inc. for a project not related to this research. The remaining authors declare no competing interests.
