## [Peer Review File · Nature Communications]

Genome-wide meta-analysis of phytosterols reveals five novel loci and a detrimental effect on coronary atherosclerosisReviewers' Comments:

Reviewer #1:

Remarks to the Author:

This analysis addresses an important clinical question since supplementation of foods with plant sterols is commonplace and widely recommended as a non-pharmacological treatment for dyslipidemia. 2 g/day of supplementation of phytosterols in food products lead to a reduction of LDL-C concentrations by 8–10%. However large population studies including EPIC-NL showed no evidence that this was associated with CV protection as would be predicted by the effects on LDL-C. This might suggest that the protective effect of lower LDL-C levels is ablated by a detrimental effect of plant sterol absorption.

The authors are to be commended for providing the first large GWAS meta-analysis for serum phytosterols and MR analysis of phytosterols and CAD risk.

They confirm and add to previously reported signals at the ABCG5/G8 and ABO loci and identify 5 novel loci HMGCR, SCARB1, APOE (E2/2 allele), NPC1L1 and PNLIPRP2.

The latter two, NPC1L1 and PNLIPRP2 as well as the new rSNP s4299376 in ABCG8 are of the greatest interest since they do not associate with plasma lipid traits.

Comments

P8 Traits were adjusted for lipid lowering medications: This is not entirely adequate unless individual statins and doses were considered. Were individuals treated with ezetimibe excluded from the analyses?

P21 Are there data showing that SR-BI increases cholesterol absorption from micelles and thus (plausibly) sitosterol absorption? Srb1 k/o mice do not have impaired cholesterol absorption J. Lipid Res. 42 (2001) 170 – 180.

P 22 Did the authors perform MR analysis using SNPs associated with sitosterols but not with cholesterol?

P23/Table 2 Please clarify “the direct effect constitutes 58%, i.e slightly larger than the indirect effect”. This sentence does not appear to accord with data in Table 2.

Figure 4: Data might be better presented for clarity. In the context of this study, suggest limit the eQTL data to the small intestine and divide the figure into phenotype associations and eQTL data. In any case, it appears that the only interesting eQTL data are for PNLIPRP2 and SCARB1. Here make the point that coding SNPs are not expected to have eQTL effects but are very important due to site of expression and functional effects.

Reviewer #2:

Remarks to the Author:

Genome-wide association meta-analysis of serum phytosterols reveals five novel loci and a detrimental causal effect on coronary artery disease risk

Scholz et al.

This manuscript describes GWAS for circulating levels of various phytosterols and their ratios to other sterols in ~9800 subjects, followed by various follow up bioinformatics analyses. The authors identify

several new loci for serum phytosterol concentrations beyond their prior study with fewer subjects, including previously unrecognized independent SNPs at the well-known ABCG5/ABCG8 locus on chromosome 2. In addition to prioritizing candidate genes at each locus, Mendelian randomization analyses provide evidence that genetically elevated sitosterol levels are causally associated with risk of CAD. This relationship was shown to be both independent of and mediated partially through total cholesterol levels. The work presented in the manuscript is a thorough evaluation of the genetic basis of serum phytosterol levels and their relationship to risk of CAD. The latter concept has been questioned in prior studies and the present analysis provides convincing evidence for a causal relationship of at least sitosterol levels with CAD. The authors may wish to address the following minor points.

1. The MR analysis should use summary statistics from a more recent GWAS meta-analysis with CARDIoGRAM+C4D and UKBB for CAD (PMID: 29212778) rather than the Nikpay paper from 2015.
2. It would also be good to determine the causal relationship between genetically elevated sitosterol levels and CAD is observed in an independent dataset of non-European ancestry subjects such as Biobank Japan (PMID: 33020668). While this would assume that effect estimates for sitosterol are the same across European and Japanese ancestry subjects, repeating the MR analyses in another population may nonetheless provide additional confirmatory evidence for a causal relationship with CAD, particularly with respect to independence from total cholesterol.
3. Line 371: Do you authors mean the PP3 for cholesterol with rs4299376 should be -99.7% (negative)?
4. ST6 and 7: Are the data in these two tables P-values or -log p-values?

POINT-BY-POINT RESPONSE TO REVIEWER COMMENTS

Reviewer #1

This analysis addresses an important clinical question since supplementation of foods with plant sterols is commonplace and widely recommended as a non-pharmacological treatment for dyslipidemia. 2 g/day of supplementation of phytosterols in food products lead to a reduction of LDL-C concentrations by 8–10%. However large population studies including EPIC-NL showed no evidence that this was associated with CV protection as would be predicted by the effects on LDL-C. This might suggest that the protective effect of lower LDL-C levels is ablated by a detrimental effect of plant sterol absorption. The authors are to be commended for providing the first large GWAS meta-analysis for serum phytosterols and MR analysis of phytosterols and CAD risk.

They confirm and add to previously reported signals at the ABCG5/G8 and ABO loci and identify 5 novel loci HMGCR, SCARB1, APOE (E2/2 allele), NPC1L1 and PNLIPRP2. The latter two, NPC1L1 and PNLIPRP2 as well as the new rSNP s4299376 in ABCG8 are of the greatest interest since they do not associate with plasma lipid traits.

Authors reply: We thank the reviewer very much for the positive evaluation and the helpful and constructive comments.

Comment 1: P8 Traits were adjusted for lipid lowering medications: This is not entirely adequate unless individual statins and doses were considered. Were individuals treated with ezetimibe excluded from the analyses?

Authors reply: We agree that our adjustment is a simplification and that consideration of dosing schemes would be more appropriate. However, this dosing information is scarcely available in population-based cohorts, e.g. we did not collect this information in our LIFE cohorts. Moreover, we did not distinguish between different pharmaceuticals but summarized all medication starting with ATC code “C10” as “lipid-lowering medication”. No exclusions with respect to medication were performed.

In the vast majority of cases, statins were prescribed (e.g. LIFE-Adult: 95%, LIFE-Heart: 97%), while only 6% respectively 5% of subjects received other drugs from the “C10” category (percentages do not add to 100% since a few subjects receive multiple medication from the “C10” category). Prescription of Ezetimib was even less frequent, with <2% in both cohorts. We expect similar percentages in the other studies due to similar or former recruitment periods.

Changes in manuscript: We added these limitations to our discussion section. We also added the definition of “lipid lowering medication” to the methods section.

Comment 2: P21 Are there data showing that SR-BI increases cholesterol absorption from micelles and thus (plausibly) sitosterol absorption? *Srb1* k/o mice do not have impaired cholesterol absorption J. Lipid Res. 42 (2001) 170 – 180.

Authors reply: We thank the reviewer very much for pointing to this publication. The scavenger receptor class B type I (SR-BI) is a receptor for high-density lipoproteins (HDL) and facilitates cholesterol delivery to steroidogenic tissues and cholesterol excretion in the liver. Indeed, there are controversial results in the literature regarding SR-BI involvement in sterol uptake.

Supporting our interpretation, there are groups showing the effect of SR-B1 on cholesterol absorption by different *in vitro* and *in vivo* approaches. In a cell culture experiment with TC7/Caco-2 cells performed by Haikal et al. (Lipids 43, 401–408 (2008)), efficient cholesterol absorption occurs from small lipid donors (≤ 23 nm diameter), mainly due to NPC1L1 and SR-BI involvement. This was shown by anti-SR-BI antibodies significantly reducing cholesterol absorption. Also, overexpression of SR-BI in

Chinese hamster ovary cells resulted in increased cholesterol uptake 1 (Altmann S et al. *Biochim Biophys Acta*. 2002 Jan 30;1580(1):77-93).

In mice, over-expressing SR-BI in intestine increased cholesterol uptake determined by radioactive acute phase measurement of plasma cholesterol as reported by Bietrix et al. (*Biol Chem* 281(11), 7214-9 (2006)). There is also evidence that upregulation of intestinal SR-BI is associated with overproduction of intestinal apoB48-containing lipoproteins (Hayashi AA et al. *Am J Physiol Gastrointest Liver Physiol*. 2011).

In contrast, the SR-BI k/o study cited by the reviewer suggests that SR-B1 is not essential for intestinal cholesterol absorption.

Changes in manuscript: We added this controversy to the discussion.

Comment 3: P 22 Did the authors perform MR analysis using SNPs associated with sitosterols but not with cholesterol?

Authors reply: The reviewer is right in pointing out that type 1 pleiotropy might be an issue for Mendelian Randomization analysis.

As primary analysis, we considered six variants as instruments for sitosterol. These variants comprise three of the four independent variants from 2p21 (ABCG5/8) and the three single variants from 7p13 (NPC1L1), 10q25.3 (PNLIPRP2) and 12q24.31 (SCARB1). The fourth independent variant of 2p21 could not be considered in Europeans due to missing CAD summary statistics in the updated data set requested by reviewer #2. All of these six SNPs showed both, genome-wide significant associations with sitosterol, and, clearly stronger effect sizes with sitosterol compared to cholesterol (see explained variances in table S9d) suggesting that these signals are driven by sitosterol rather than cholesterol association. We also performed a sensitivity analysis by using only the instruments from the 2p21 locus for which the functional role in phytosterol excretion is well established.

As instruments for cholesterol, we considered up to 36 variants excluding those showing sitosterol associations. This was ensured by excluding cytobands with phytosterol associations. In a sensitivity analysis, we considered only the strongest cholesterol associations as instruments.

All sensitivity analyses showed similar results (Table S14). Moreover, methods more robust to pleiotropy issues (such as MR-Egger) also yielded similar results (supplementary figures S10). By these measures, we aimed at avoiding pleiotropy driven causal effects as far as possible.

Changes in manuscript: We improved the discussion of the pleiotropy issue. We also improved the presentation of the instruments used for Mendelian Randomization in Methods and Results (new supplemental tables S16 and S17). Mendelian Randomization was repeated using more recent summary statistics of coronary artery disease as requested by reviewer #2 (see comment 1). We also added results of Biobank Japan as requested by reviewer #2 (see comment 2). New results are presented in revised table 2, revised supplemental table S14 and new supplemental table S15.

Comment 4: P23/Table 2 Please clarify “the direct effect constitutes 58%, i.e slightly larger than the indirect effect”. This sentence does not appear to accord with data in Table 2.

Authors reply: We regret that our statement might be misleading. We aimed to compare the relative sizes of the causal direct and indirect effect estimates shown in table 2. The direct effect was 0.213 while the indirect effect was 0.154. The sum of both constitute the total effect (0.367), i.e. the direct effect was 58% of the total effect.

Changes in manuscript: Since we repeated the Mendelian Randomization analyses using more recent CAD summary statistics as requested by reviewer #2, all numbers in table 2 where changed. We also rewrote the statement regarding comparisons of direct and indirect effects hoping that it is clearer now.

Comment 5: Figure 4: Data might be better presented for clarity. In the context of this study, suggest

limit the eQTL data to the small intestine and divide the figure into phenotype associations and eQTL data. In any case, it appears that the only interesting eQTL data are for PNLIPRP2 and SCARB1. Here make the point that coding SNPs are not expected to have eQTL effects but are very important due to site of expression and functional effects.

Authors reply: We thank the reviewer very much for the suggestion. Indeed, eQTLs in small intestine are probably the most relevant results. However, since gene-expression of ABCG8 in colon tissue is also described including strong eQTLs, we prefer to show respective colocalization results too. We believe that it might be of interest for the reader that some of the 2p21 hits colocalize with eQTLs but others not. This could provide functional explanations for the observed independent genetic effects at this locus.

Changes in manuscript: We adapted the figure, i.e. we removed the eQTLs of blood and pancreas as suggest by the reviewer but keep the results for colon. We also separate eQTL results from those of the phenotypes. We also mention in the methods section that colocalization is not a necessary condition of functional plausibility of a locus.

Reviewer #2:

Genome-wide association meta-analysis of serum phytosterols reveals five novel loci and a detrimental causal effect on coronary artery disease risk

Scholz et al.

This manuscript describes GWAS for circulating levels of various phytosterols and their ratios to other sterols in ~9800 subjects, followed by various follow up bioinformatics analyses. The authors identify several new loci for serum phytosterol concentrations beyond their prior study with fewer subjects, including previously unrecognized independent SNPs at the well-known ABCG5/ABCG8 locus on chromosome 2. In addition to prioritizing candidate genes at each locus, Mendelian randomization analyses provide evidence that genetically elevated sitosterol levels are causally associated with risk of CAD. This relationship was shown to be both independent of and mediated partially through total cholesterol levels. The work presented in the manuscript is a thorough evaluation of the genetic basis of serum phytosterol levels and their relationship to risk of CAD. The latter concept has been questioned in prior studies and the present analysis provides convincing evidence for a causal relationship of at least sitosterol levels with CAD. The authors may wish to address the following minor points.

Authors reply: We thank the reviewer very much for the positive evaluation and the helpful and constructive comments.

Comment 1: The MR analysis should use summary statistics from a more recent GWAS meta-analysis with CARDIoGRAM+C4D and UKBB for CAD (PMID: 29212778) rather than the Nikpay paper from 2015.

Authors reply: We thank the reviewer very much for this suggestion. Indeed, there are more recent summary statistics for CAD available. We therefore replaced all of our Mendelian randomization analysis using this new data set. The results were very similar, i.e. all major findings were preserved.

Changes in manuscript: We replaced the Mendelian randomization analyses of Europeans. We adapted table 2, supplemental table S14 and supplemental figure S10 accordingly and rewrote the respective parts of the paper to account for the new results.

Comment 2: It would also be good to determine the causal relationship between genetically elevated sitosterol levels and CAD is observed in an independent dataset of non-European ancestry subjects such as Biobank Japan (PMID: 33020668). While this would assume that effect estimates for sitosterol are the same across European and Japanese ancestry subjects, repeating the MR analyses in another population may nonetheless provide additional confirmatory evidence for a causal relationship with CAD, particularly with respect to independence from total cholesterol.

Authors reply: Indeed, it would be interesting to analyse whether the observed causal relationships can be translated to another ethnicity. As pointed out by the reviewer, we have to assume that genetic sterol associations are the same in this ethnicity, i.e. that instrumental variable effects can be translated. The results of this analysis were very similar to those obtained for Europeans. However, these results should be considered with caution since we observed larger differences in allele frequencies of instruments between European and Japanese ethnicities.

Changes in manuscript: We added the requested analysis to the revised manuscript. Results are shown in new supplemental table S15 and new supplemental figure S11. We also mention the analysis in our methods and results section and critically discussed the results.

Comment 3: Line 371: Do you authors mean the PP3 for cholesterol with rs4299376 should be -99.7% (negative)?

Authors reply: We regret that the presentation of posterior probabilities might be misleading. Actually, posterior probabilities of H0 to H4 are defined as positive and add up to 100%. In this regard, the

scaling used in figure 4 might be misleading since the posterior probabilities for H3 were presented as negative values.

Changes in manuscript: We explained the intended normalization of the posterior probabilities in the methods section. We also improved the legends of figures 4 and S8 hoping that this is clearer now.

Comment 4: ST6 and 7: Are the data in these two tables P-values or -log p-values?

Authors reply: In these tables, $-\log_{10}$ of p-values are presented. We regret that this was not mentioned in the previous version of the tables.

Changes in manuscript: We added this information in the table legends.

Reviewers' Comments:

Reviewer #1:

Remarks to the Author:

No further comments

Reviewer #2:

Remarks to the Author:

The revised manuscript by Scholz et al has addressed the point I raised in my initial review. I also noted that they have been responsive to Reviewer 1 as well. I only made some additional minor suggestions regarding some supplemental figures after which the manuscript would be suitable for publication from my point of view.

Best,
Hooman